# Effects of Cinnamon Essential Oil on Oxidative Damage and Outer Membrane Protein Genes of *Salmonella enteritidis* Cells

**DOI:** 10.3390/foods11152234

**Published:** 2022-07-27

**Authors:** Zhen Zhang, Yuanyuan Zhao, Xueqin Chen, Wei Li, Wen Li, Jianming Du, Li Wang

**Affiliations:** College of Food Science and Engineering, Gansu Agricultural University, Lanzhou 730070, China; 18119300298@163.com (Y.Z.); 18409447558@163.com (X.C.); weili@gsau.edu.cn (W.L.); lwengsau@163.com (W.L.); my17339830698@163.com (J.D.); m15294207998@163.com (L.W.)

**Keywords:** cinnamon essential oil, *Salmonella enteritidis*, oxidative damage, outer membrane protein

## Abstract

Salmonella is an important pathogen causing food poisoning. Food safety and health are the themes of today′s society. As a class of food-borne pathogens, *Salmonella enteritidis* had become one of the common zoonotic pathogens. Cinnamon essential oil (CEO) had been reported as an antibacterial agent, but there are few studies on its antibacterial mechanism. This study investigated the effects of CEO on oxidative damage and outer membrane protein genes of *Salmonella enteritidis* cells. First, the reactive oxygen species content in bacteria treated with different concentrations of cinnamon essential oil was determined by fluorescence spectrophotometry, and the effects of superoxide dismutase (SOD), catalase (CAT) and superoxide dismutase (SOD), and catalase (CAT) and peroxidase (POD) were determined by the kit method. The activity of POD and the content of malondialdehyde (MDA) were investigated to investigate the oxidative damage of CEO to *Salmonella enteritidis* cells. By analyzing the effect of CEO on the *Salmonella enteritidis* cell membrane’s outer membrane protein gene expression, the mechanism of CEO′s action on the *Salmonella enteritidis* cell membrane was preliminarily discussed. The results showed that CEO treatment had an obvious oxidative damaging effect on *Salmonella enteritidis*. Compared with the control group, the increase in CEO concentration caused a significant increase in the bacteria ROS content. The observation technique experiment found that with the increase in CEO concentration, the number of stained cells increased, which indicated that CEO treatment would increase the ROS level in the cells, and it would also increase with the increase in CEO concentration, thus causing the oxidation of cells and damage. In addition, CEO treatment also caused the disruption of the balance of the cellular antioxidant enzymes (SOD, CAT, POD) system, resulting in an increase in the content of MDA, a membrane lipid metabolite, and increased protein carbonylation, which ultimately inhibited the growth of *Salmonella enteritidis*. The measurement results of cell membrane protein gene expression levels showed that the Omp genes to be detected in *Salmonella enteritidis* were all positive, which indicated that *Salmonella enteritidis* carried these four genes. Compared with the control group, the relative expressions of OmpF, OmpA and OmpX in the CEO treatment group were significantly increased (*p* < 0.05), which proved that the cell function was disturbed. Therefore, the toxicity of CEO to *Salmonella enteritidis* could be attributed to the damage of the cell membrane and the induction of oxidative stress at the same time. It was speculated that the antibacterial mechanism of CEO was the result of multiple effects. This work was expected to provide a theoretical basis for the development of new natural food preservatives and the prevention and control of *Salmonella enteritidis*.

## 1. Introduction

With the continuous increase in the world′s population, the demand for food types had gradually increased, and more and more food safety issues have been brought to the forefront. Therefore, food safety issues has become one of the important challenges facing today′s society. According to the World Health Organization report, Salmonella was at the forefront of food-borne pathogens. Salmonella contamination of different degrees exists in most parts of the world, which seriously threatens public health and has caused serious economic losses [1]. Salmonella belongs to enterobacteriaceae and is a Gram-negative microorganism with flagella, no spores, and red after staining [2]. Although chemical preservatives have been widely developed and used, they had brought great harm to human health. Therefore, it was of great significance to choose and develop green and natural antibacterial substances.

At present, spice essential oil (EO) is widely used in food preservation, and its composition is complex [3]. Studies had shown that EOS had obvious inhibitory effects on important food-borne pathogens such as salmonella [4,5]. Cinnamon is a dicotyledonous Lauraceae plant, and its bark, leaves and other tissues contain a large amount of volatile oils, flavanols, polyphenols, coumarin, lignin, flavonoids, etc. [6]. Cinnamon essential oil (CEO) was extracted from cinnamon bark, cinnamon leaves, etc. It was a yellow oily liquid that is non-toxic and volatile. Studies have shown that the main component of CEO was cinnamaldehyde [7,8]. Negar Mortazavi′s study found that the combined effect of ultrasound (US) and CEO could effectively inhibit the growth of *Listeria monocytogenes* and *Salmonella typhimurium* in low-fat and high-fat milk, but the cause was not explored [9]. Mariem Somrani et al. could effectively inhibit the adhesion of initial cells of *Listeria monocytogenes* on biofilms [10]. Jeong found that CEO could effectively inhibit the growth of oral microorganisms, including acid bacteria that cause dental caries [11]. He found that CEO could destroy the cell membrane integrity of *Clostridium oxysporum*, which is the main pathogen causing “Hongyang” kiwifruit anthracnose [12]. Elcocks and other studies found that CEO could quickly kill *Pseudomonas aeruginosa* by affecting the permeability and integrity of its cell membrane [13]. However, there were few reports on CEO′s research on the mechanism of salmonella cell damage.

The chemical properties of ROS free radicals were very active. They were generated by aerobic metabolism in the body or stimulated by external factors, and their accumulation would damage biological macromolecules such as proteins in the body [14,15]. Originally, the generation and scavenging of ROS was a dynamic equilibrium process, but when microorganisms were subjected to stress such as bacteriostatic substances, the content of ROS in the body would be greatly increased, and SOD, CAT and POD in the protective enzyme system could be responsible for scavenging it [16,17]. However, too much ROS would cause the bacteria to exceed its own antioxidant capacity, resulting in this balance being broken, and cell components such as lipids, proteins, and nucleic acids in the body would be irreversibly damaged under the action of ROS [18,19]. Therefore, this paper intended to use CEO as the antibacterial substance and *Salmonella enteritidis* as the test bacteria to study the effect of CEO on the cellular oxidative damage of *Salmonella enteritidis*. The main contents included the determination of ROS content, MDA content and antioxidant enzyme activities in *Salmonella enteritidis* after CEO treatment, and the changes of ROS content were observed by fluorescence microscope. The purpose was to explore the effect of CEO treatment on ROS generation and the scavenging system in *Salmonella enteritidis*. In order to evaluate the degree of oxidative damage to the bacterial cell membrane by CEO, it was of great theoretical significance to deeply understand the antibacterial mechanism of CEO on *Salmonella enteritidis*. Omp content accounts for about half of the outer membrane structure of Gram-negative bacteria, and they polymerized with peptidoglycan to form a channel for the entry and exit of substances [20,21]. Omp was not only the structural and functional protein of bacteria but also an important virulence factor of bacteria. Omp played an important role in participating in the material metabolism of the bacteria, such as the transportation of materials (including controlling the important channels for antimicrobial drugs to enter the bacteria); it maintained the normal shape of bacteria (supporting the outer cell membrane skeleton) and regulated the synthesis of related substances [22]. Studies had confirmed that antibacterial drugs were blocked from the target by Omp channels (Omp could restrict the entry of antibacterial drugs into cells by controlling the pore size of the channel) before it entered the bacteria, and it could also form an efflux system to remove the antibacterial drugs that have entered the cell. It was pumped out of the cell, but some antibacterial drugs could achieve an antibacterial effect by increasing the permeability of Omp [23,24]. When a certain Omp function was inhibited, the bacteria would activate the expression of other Omp for their own survival. On this basis, this chapter further analyzed the expression changes of Omp genes (OmpA, OmpF, OmpW and OmpX) from the gene level, and it further improved the inhibitory mechanism of CEO on *Salmonella enteritidis* cell membrane, in order to clarify the antibacterial effect of CEO on *Salmonella enteritidis*. The mechanism provided a certain theoretical basis for the development of safe and efficient *Salmonella enteritidis* inhibitors.

In this study, by measuring protein carbonylation levels as well as ROS, MDA, SOD, CAT and POD content, we evaluate the damage effect and mechanism of CEO on *Salmonella enteritidis* and further study its damage mechanism from OmpA and OmpX, OmpW, and OmpF gene levels. This work was expected to provide a theoretical basis for the development of new natural food preservatives and the prevention and control of *Salmonella enteritidis*.

## 2. Materials and Methods 

### 2.1. Materials 

The cinnamon used in the test was purchased from Baiweifu Food Co., Ltd., from Luoding, Guangdong, China. The experimental strain *Salmonella enteritidis* (ATCC BAA-664) was purchased from the China Microbial Culture Collection Center. We weighed an appropriate amount of cinnamon bark into a pulverizer to pulverize and then passed it through a 40-mesh sieve to collect the powder for later use. Then, we weighted the appropriate amount of cinnamon peel and put it into a grinder to crush. Afterwards, we passed it through a 40-mesh sieve, collected the powder and set it aside. CEO extraction with reference to MarjanaRadünz′s method used 200 g of cinnamon powder in a 3000 mL round-bottom flask, to which we added 2000 mL of distilled water. Then, we shook it thoroughly, heated it at a micro-boil for the start point timing and subjected it to distillation for 4 h to obtain the CEO [25].

### 2.2. Antioxidant Activity Assay

#### 2.2.1. Determination of ROS

The Nanjing Jiancheng ROS assay kit was used to determine the ROS content of the samples. Different concentrations (1/2 MIC, MIC) of CEO were added to the bacterial suspensions cultured to the logarithmic growth phase for different time periods. MIC was 0.8 μL·mL^−1^, and 1/2 MIC was 0.4 μL·mL^−1^, which was obtained from previous experience. Samples were taken at 0, 2, 4, 6, and 8 h of incubation at 150 r/min. Then, they were centrifuged, the supernatant was discarded, and they were washed three times with PBS to obtain the pellet for later use. The reaction was carried out at 37 °C for 60 min, and the mixture was mixed once for 5 min. Then, we centrifuged them at 888 g for 5 min, washed them twice with PBS to obtain the pellet, resuspended them, and finally measured the fluorescence intensity.

#### 2.2.2. O_2_^−^ Free Radical Determination

We took the bacterial suspension cultured to the logarithmic growth phase, added CEO with different concentrations (1/2 MIC, MIC) for different times, set up the control group at the same time, and cultivated this suspension at 37 °C and 150 r/min for 0, 2, and 4 h. Sampling took place at 6 and 8 h in a shaker, after which it was centrifuged at 12,000× *g* for 10 min at 4 °C. Then, we discarded the supernatant and resuspended it in 4 mL of PBS. The supernatant was collected by centrifugation and the subsequent experiments were completed within 50 min. Referring to the kit operation, we finally determined the OD_530_ [26].

#### 2.2.3. Fluorescence Microscopy to Observe Changes in ROS Content

Referring to the method of Chiesa [27] with slight modifications, firstly, the bacteria cultured to the logarithmic phase were treated with different concentrations (1/2 MIC and MIC) of CEO at 37 °C for 4 h, and a control group was set. After centrifuging at 5000× g for 5 min, we collected the precipitate, washed it three times with PBS and resuspended it. Then, we took 200 μL of bacterial solution, mixed it with 20 μM 2′,7′-dichlorodihydrofluorescein diacetate and incubated it for 1 h, mixing once every 5 min. Then, we washed and resuspended it with PBS and used fluorescence microscopy to record the results.

### 2.3. Enzyme Activity Determination

#### 2.3.1. SOD Activity Determination

Referring to the method of Tan and Ramazani et al., we used the SOD kit to measure the SOD activity of the sample [28,29]. First, the bacterial suspension cultured to the logarithmic growth phase was added with different concentrations (1/2 MIC, MIC) of CEO to treatment for 0, 2, 4, 6, and 8 h. Then, the pellets were subject to centrifugation, washed with PBS and resuspended, and then sonicated, and centrifuged at 4 °C to keep the supernatant. According to the SOD kit, the OD_450_ was finally determined. Each experiment was repeated three times, and the average value was calculated.

#### 2.3.2. CAT, POD and MDA Activity Determination

The pretreatment was the same as the SOD activity determination, and then, the operation was performed with reference to the CAT kit, POD kit and MDA kit. Finally, the microplate reader was read at 405 nm, 420 nm, and 532 nm. Each experiment was repeated three times, and the results were averaged [30,31].

### 2.4. Determination of Protein Carbonylation

Referring to the method of Mohammad MehdiOmmati et al. [32] with slight modifications, a 1/2 MIC and MIC concentration of CEO were added to the bacterial suspension cultured to the logarithmic phase, which was then treated at 37 °C and 150 r/min. Samples were taken at 0, 2, 4, 6, and 8 h; the precipitate was collected by centrifugation, washed with physiological saline, and 10 mL of lysis buffer was added (50 μL of physiological saline, 10 μL of 0.1 mol.L-1 PMSF, pH 8.0 Tris-HCl 940 μL). After sonication (300 W, intermittent 2 s, total time 5 min) and centrifugation at 15,000 r/min for 30 min, the supernatant was retained for the determination of carbonylation content. Then, 400 μL of 2,4-dinitrophenylhydrazine (DNPH) was added to 100 μL of supernatant, and a blank control group was set up for 1 h in the dark. Then, we added 2 mL of trichloroacetic acid and centrifuged at 12,000× *g* for 20 min at 4 °C to obtain a precipitate. We washed three times with ethyl acetate and ethanol solution, added 1.25 mL of guanidine hydrochloride, reacted at 37 °C for 15 min, and then centrifuged the supernatant to measure OD_370_.

### 2.5. The Effect of CEO on Omp Gene Transcription Level

#### 2.5.1. PCR Amplification

The DNA of *Salmonella enteritidis* was extracted with bacterial genomic DNA extraction kit as the template, and primers were added to amplify the target gene fragment [33].

#### 2.5.2. Agarose Gel Electrophoresis

First, we prepared a 3% agarose gel solution, prepared a spotting gel block, and added the sample to the spotting plate at a ratio of 6:1 (DNA sample: 6 × Loading Buffer). Then, for electrophoresis, we set the voltage to 60–100 V and the time to 30–45 min. After the electrophoresis was over, we used a gel imaging system to take pictures and recorded the results [34].

#### 2.5.3. Preparation of *Salmonella enteritidis* cDNA

*Salmonella enteritidis* bacterial suspension in logarithmic growth phase was added to CEO at different concentrations (1/2 MIC, MIC), and a control group was set up, cultured at 37 °C for 4 h, washed twice with normal saline, and centrifuged to collect the precipitate for later use. The total RNA was extracted according to the operation of the bacterial total RNA rapid extraction kit, and the cDNA was obtained by reverse transcription and finally used for q-PCR test.

#### 2.5.4. q-PCR Test

Total RNA was extracted according to the operation of the bacterial total RNA rapid extraction kit, and cDNA was obtained by reverse transcription. The 16s rRNA was used as the internal reference gene for the q-PCR test. Primer designs for q-PCR are shown in Appendix A.

### 2.6. Data Analysis

Data processing was performed using Microsoft Excel 2013; statistical analysis was performed using SPSS Statistics 25.0 software (*p* < 0.05); plotting was performed by Origin 2018 software.

## 3. Results and Analysis

### 3.1. Antioxidant Activity Results

#### 3.1.1. Determination of ROS

ROS had certain effects on multiple life processes such as cell growth and death [35]. Therefore, this study analyzed the effect of CEO on the oxidative damage of *Salmonella enteritidis* by measuring the changes of intracellular ROS content in *S. enteritidis*. As shown in Figure 1, the content of ROS in the treatment group was significantly higher than that in the control group (*p* < 0.05). Among them, in the 1/2 MIC and MIC treatment groups, the relative fluorescence intensity of ROS increased with the prolongation of treatment time. However, the increase of 1/2 MIC was not as large as that in the MIC treatment group. The results showed that the CEO treatment of *Salmonella enteritidis* could lead to the increase in the ROS content in the cells, and it showed a CEO concentration dependence: that is, the higher the CEO concentration, the higher the ROS content.

#### 3.1.2. Determination of O_2_^−^ Free Radicals

ROS mainly existed in the form of O_2_^−^, which is ubiquitous in microorganisms, has strong oxidizing properties and plays an important role in the metabolic process in vivo [36]. When the antibacterial agent acted on the bacteria, the ROS content in the bacteria would increase, but when the increased amount exceeds the antioxidant defense ability of the bacteria itself, the protein and other intracellular substances in the bacteria will be irreversibly damaged, which will interfere with various metabolic processes of cells [37]. Figure 2 reflected the change of O_2_^−^ content with treatment time. The content of O_2_^−^ in the CEO treatment group showed an increasing trend. Moreover, the O_2_^−^ content of the treatment group was significantly greater than that of the control group (*p* < 0.05), and it showed a CEO concentration dependence: that is, the greater the CEO concentration, the higher the O_2_^−^ content, which indicated that CEO treatment induced intracellular bacteria. The accumulation of O_2_^−^ causes oxidative damaged to cells and inhibited the growth of bacterial cells. 

#### 3.1.3. Fluorescence Microscopy to Observe the Changes of ROS

ROS was an aerobic metabolite in the body, which could oxidize 2′,7′-dichlorodihydrofluorescein diacetate to DCF, and DCF had green fluorescence [38]. A small amount of ROS was beneficial to the normal progress of intracellular metabolism, but a large amount of accumulated ROS would exceed the antioxidant capacity of the bacteria. That was, normal cells would also produce a small amount of ROS to maintain the normal process of intracellular metabolic activities, but excessive ROS would cause serious damage to the bacteria [39,40]. As shown in Figure 3, almost no fluorescence was produced in the control group, and a small amount of fluorescence appeared in the 1/2 MIC treatment group; meanwhile, a large amount of fluorescence was produced in the MIC and 2 MIC treatment groups, indicating that CEO treatment would increase the level of intracellular ROS, and as the concentration of CEO increases, it would thereby cause oxidative damage to cells.

#### 3.1.4. SOD Activity Assay

SOD was one of the key protective enzymes in bacteria, which could scavenge O_2_^−^ and improve the tolerance of bacteria to ROS so as to protect the cells from oxidative damage [41]. As shown in Figure 4, the results showed that after *Salmonella enteritidis* was treated with different concentrations of CEO for 2 h, the SOD activity decreased to varying degrees. At 4 h, the SOD activity increased rapidly, and the higher the CEO concentration, the greater the increase. Among them, the SOD activity of the MIC treatment group reached the maximum at 4 h, which was 149.16 U·mgprot^−1^; at 8 h, the SOD activity decreased significantly, and the higher the CEO concentration, the more significant the SOD activity decreased (*p* < 0.05), while the control group showed no significant change. Therefore, it was shown that CEO treatment had a significant effect on the SOD activity of *S. enteritidis*, and the SOD activity decreased, indicating that its protection to *Salmonella enteritidis* itself decreased, and it was easily attacked by free radicals and autolyzed.

#### 3.1.5. CAT Activity Assay

CAT was one of the important protective enzymes for scavenging ROS in bacteria, which could disproportionate H_2_O_2_ in the body to generate H_2_O and O_2_, so that the bacteria cells could be protected from the destruction of H_2_O_2_ [42]. As shown in Figure 5, after *Salmonella enteritidis* was treated with 1/2 MIC and MIC concentration CEO for 4 h, the CAT activity of *Salmonella enteritidis* showed a significant upward trend (*p* < 0.05) and reached the maximum value of 87.37 U·mgprot^−1^ and 97.74 U·mgprot^−1^ at 4 h and 8 h, respectively. The CAT activity in the treatment group decreased significantly, and the higher the CEO concentration, the more obvious the decrease, while the control group had no significant change. The above results indicated that the CAT activity of *Salmonella enteritidis* was significantly reduced under CEO stress. The increase in CAT enzyme activity in the early stage may be to protect the cells, but in the later stage, the cells senesce, the metabolic intensity decreases, and the CAT activity decreases, which resulted in a reduced ability to scavenge ROS.

#### 3.1.6. POD Activity Assay

POD could scavenge excess free radicals, and the reduction in its activity would promote the oxidative damage of the bacteria, thereby hindering the growth and metabolism [43]. Figure 6 showed the results of bacterial POD activity after treatment with different concentrations of CEO. Compared with the control group, the POD activity of the CEO treatment group was significantly decreased (*p* < 0.05), and showed a concentration-dependent manner, that was, the higher the CEO concentration, the lower the POD activity. The POD activity of the group decreased by 43.92%; the MIC treated group decreased by 59.48%. It could be seen that CEO could significantly inhibit the POD activity of the bacteria, thereby affecting its ability to scavenge free radicals.

#### 3.1.7. Determination of MDA Content

As the final product of membrane lipid peroxidation, MDA was an important indicator to measure the degree of oxidative damage in bacterial cells [44]. Free radicals could cause membrane lipid peroxidation to produce MDA and cause oxidative damage to cells. As shown in Figure 7, the MDA content of *Salmonella enteritidis* was significantly increased after 2 h of treatment with different concentrations of CEO; the growth rate of MDA content slowed down at 4 h, which might be due to the accumulation of free radicals caused by the addition of CEO in the early stage, and membrane lipids were affected by the attack of free radicals, which increased the MDA content. With the prolongation of the CEO induction time, the SOD activity increased, and a large number of free radicals were scavenged, so the MDA changes tended to be gentle; at 8 h, the MDA content of the 1/2 MIC and MIC treatment groups were the same. They significantly increased, 1.93 and 3.51 nmol·mgprot^−1^, respectively, and the greater the CEO concentration, the more significant the increase in MDA content (*p* < 0.05). This result indicated that CEO could increase MDA content by inhibiting the activity of antioxidant enzymes in *Salmonella enteritidis* cells.

### 3.2. Determination of Protein Carbonylation

Proteins play an important role in various life activities. The degree of damage to proteins by free radicals was no less than that of lipids. After oxidative damage to proteins, amino acids would be carbonylated, and catalytic enzymes would lose their activity [42]. However, the carbonylation of proteins could not be repaired, resulting in the reduction or disappearance of their functions. The burden of its hydrolase increases, causing imbalance, so protein carbonyl content was an important indicator to measure its oxidative damage [45,46]. Therefore, in this experiment, the protein oxidative damage degree of *Salmonella enteritidis* was further speculated by measuring the protein carbonylation content of CEO on *S. enteritidis*. It could be seen from Figure 8 that compared with the control group, the carbonylation content of the CEO treatment group was significantly increased (*p* < 0.05). After 8 h of treatment, compared with the control group, the protein carbonylation content of the 1/2 MIC treatment group increased. The protein carbonylation content of the MIC treatment group increased by 92.0%. It showed a concentration dependence of CEO, that was, the protein carbonylation content of the 1/2 MIC-treated group was lower than that of the MIC-treated group. It could be seen that CEO treatment increased the degree of protein carbonylation in *Salmonella enteritidis*, resulting in increased protein oxidation, thereby affecting the life activities of bacteria and inhibiting growth.

### 3.3. The EFFECT of CEO on Omp Gene Transcription Level

CEO toxicity to *Salmonella enteritidis* could be attributed to membrane disruption and oxidative stress-induced cell inactivation or death. The Omp of *Salmonella enteritidis* was involved in the metabolism of bacterial substances, such as maintaining cell membrane fluidity, cell structure, ensuring material transport, etc. It played an important role, among which OmpA, OmpF, OmpW and OmpX were the coding genes of the four Omps of *S. enteritidis*, respectively. OmpW and OmpF were related to the transport of substances in the cell membrane; OmpA was related to the maintenance of outer membrane integrity and cell morphology; and OmpX overexpression would reduce the sensitivity of bacteria to certain drugs and lead to the development of drug resistance [47].

Therefore, based on the effect of CEO on DNA, this experiment further analyzed its effect on the expression levels of the four Omp genes. First, the Omp gene of *Salmonella enteritidis* was verified. As shown in Figure 9, the Omp genes to be detected in *Salmonella enteritidis* were all positive, which indicated that *Salmonella enteritidis* carried these four genes. On this basis, the effect of CEO on the transcription levels of the four Omp genes was further analyzed by q-PCR. Compared with the control group, the relative expression levels of OmpF in the 1/2 MIC and MIC treatment groups were significantly increased (*p* < 0.05). The relative expression level of OmpW was not significantly different in the 1/2 MIC-treated group, but it was significantly increased in the MIC-treated group, which proved that the cell function was disturbed. Compared with the control group, the relative expression levels of OmpA in the 1/2 MIC and MIC treatment groups were up-regulated by 93.05% and 97.94%, respectively; while the relative expression levels of OmpX were up-regulated by 86.43% and 96.99%, respectively, compared with the control group. That was, CEO significantly increased the expression levels of OmpA and OmpX, which might be due to the activation of the defense mechanism of *Salmonella enteritidis*. The above results indicated that the expression of the Omp gene affecting *Salmonella enteritidis* was an important target of CEO′s antibacterial effect.

## 4. Discussion

Peroxidative damage to cells under adverse conditions was one of the causes of bacterial cell death [48]. The content of ROS generated and scavenged in normal cells maintains a dynamic balance, but when the cells were stressed by antibacterial agents, ROS would accumulate in large quantities, the enzymatic activity of scavenging ROS in the cells would decrease, and the dynamic balance would be disrupted [49]. Many studies had also found that ROS could participate in the bacteriostatic effect of antibacterial agents: that was, the presence of antibacterial agents would generate a large amount of ROS, which would then react with biological macromolecules such as proteins, resulting in oxidative damage to cells [50,51].

In the experiments of this paper, the changes of ROS content were observed by measurement and fluorescence microscope. It was found that a large amount of ROS would be produced in *Salmonella enteritidis* cells treated with CEO, with the increase in CEO concentration indicating that CEO could induce bacterial cells to produce a large amount of ROS [52]. It also caused oxidative damage to cells, thereby playing a bacteriostatic effect [53,54]. At the same time, it was found that the activities of the protective enzymes SOD, CAT and POD decreased, resulting in the accumulation of ROS in the bacteria that could not be cleaned up in time, and the content of MDA increased. This may be the CEO′s stress on *Salmonella enteritidis*, which aggravated the degree of cell peroxidation [55]. As for whether CEO treatment induced the activity of antioxidant enzymes after oxidative stress or directly inhibited the activity of antioxidant enzymes, the accumulation of ROS resulted in cellular oxidative damage, or both, which remain to be further studied [56]. Ji and other studies had shown that the important reason for the oxidative damage of bacteria was the accumulation of ROS, and the activity of free radical scavenging enzymes in cells decreased, resulting in aggravated oxidative damage, increased MDA content and metabolic disorder [57,58]. Zheng found that after treatment of Candida albicans with vanilla essential oil, a large amount of ROS accumulated in cells, which in turn enhanced the degree of lipid peroxidation and then produced a large amount of MDA, which was toxic to cells [59]. Cao et al. found that citral treatment would increase the degree of lipid peroxidation in the cell membrane of Vibrio parahaemolyticus, which was consistent with the results of this experimental study [60].

Whether CEO induced cell apoptosis through other pathways was further explored in this study. Omp content accounts for about half of the outer membrane structure of Gram-negative bacteria, and they were polymerized with peptidoglycan to form a channel for the entry and exit of substances [20,21]. Omp was not only the structural and functional protein of bacteria but also an important virulence factor of bacteria. Omp played an important role in participating in the material metabolism of the bacteria, such as the transportation of materials (including controlling the important channels for antimicrobial drugs to enter the bacteria), maintained the normal shape of bacteria (supporting the outer cell membrane skeleton) and regulated the synthesis of related substances [22]. Studies had confirmed that antibacterial drugs were blocked from the target by Omp channels (Omp could restrict the entry of antibacterial drugs into cells by controlling the pore size of the channel) before they entered the bacteria, and it could also form an efflux system to remove the antibacterial drugs that had entered the cell. It was pumped out of the cell, but some antibacterial drugs could achieve an antibacterial effect by increasing the permeability of Omp [23,24]. When a certain Omp function was inhibited, the bacteria would activate the expression of other Omp for their own survival. It had been found that OmpA was a transmembrane protein, which played a key role in the stability of the membrane structure and normal metabolism; the expression of OmpX could cause the expression of some major pore proteins to decrease, which in turn led to the generation of bacterial resistance. In this study, the expression of OmpA and OmpX increased after CEO treatment, which might be due to the activation of the defense mechanism of *Salmonella enteritidis*; OmpW was related to bacterial virulence and pathogenicity, and its increased expression could enhance bacterial virulence. OmpW was also a transport channel for hydrogen peroxide and hypochlorous acid, and its decreased expression would protect bacteria from free radical damage [61]. Making the bacteria susceptible to free radical damage, OmpF was the main pore protein on the outer membrane of the cell. Due to the large pore radius of OmpF, it was easier for toxic substances to pass through [62]. OmpF gene deletion could significantly enhance the resistance of the strain to antibiotics [63]; this study found that its expression increased after CEO treatment, making it easier for CEO to cause damage to the bacteria through this channel.

The use of cinnamon extracts and essential oils could be beneficial for human health and considered as an alternative agent for antimicrobial therapy, medical applications, and antibacterial supplement in health products. In addition, no acute or chronic toxicity, no mutagenicity or genotoxicity, and no carcinogenicity have been detected in mammalian studies [64]. Wijesinghe et al. evaluated the toxicity of Cinnamomum cassia leaf EO to host tissues by using an in vitro cell culture model of the human non-cancer keratinocytes (HaCaT) cell line. The results showed that the safe use of Cinnamomum cassia leaf EO up to 1000 mg/mL had no toxic effect on human cells [65].

## 5. Conclusions

Our study elucidates that the damage mechanism of CEO on *Salmonella enteritidis* cells. CEO treatment could induce Salmonella Enteritidis to produce a large amount of ROS, MDA and protein carbonylation levels, which could cause oxidative damage to cells and irreversible damage to cells. At the same time, CEO could inhibit the activity of bacterial protective enzymes SOD, CAT and POD, which weakened the self-protection and defense function of bacteria and created excess *Salmonella enteritidis* cells; in addition, the free radicals of *Salmonella enteritidis* could not be removed, causing serious damage to cells and thereby achieving a bacteriostasis effect. CEO treatment could increase the expression of OmpA, OmpX, OmpW, and OmpF genes to disrupt the normal metabolism of the bacteria and achieve a bacteriostatic effect.

## Figures and Tables

**Figure 1 foods-11-02234-f001:**
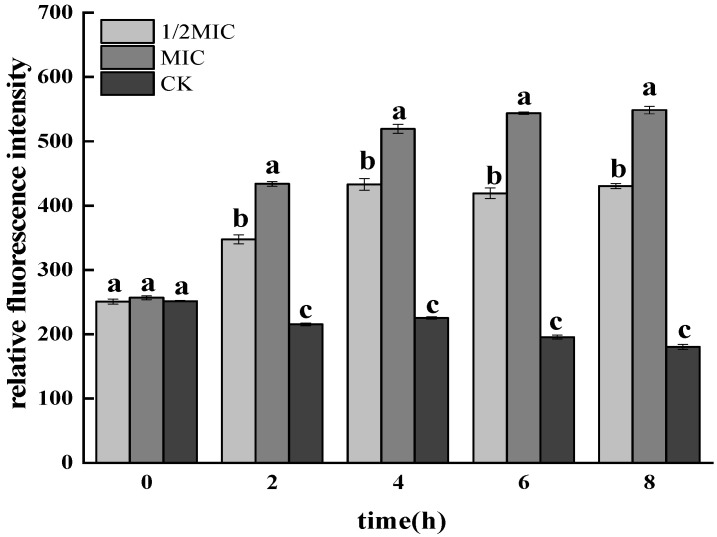
Effects of CEO treatment on intracellular ROS in *S. enteritidis.* (note: a–c indicate significant differences (*p* < 0.05) among 1/2 MIC, MIC, 2 MIC treatments).

**Figure 2 foods-11-02234-f002:**
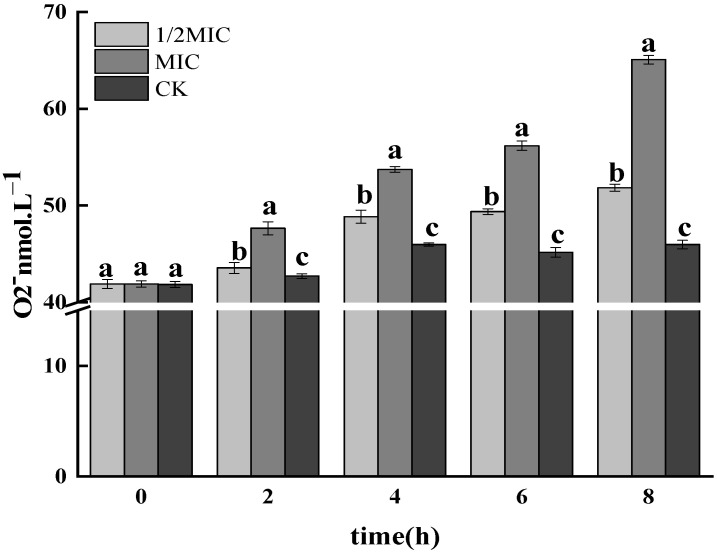
Effects of CEO treatment on intracellular O_2_^−^·in *S. enteritidis.* (note: a–c indicate significant differences (*p* < 0.05) among 1/2 MIC, MIC, 2 MIC treatments).

**Figure 3 foods-11-02234-f003:**
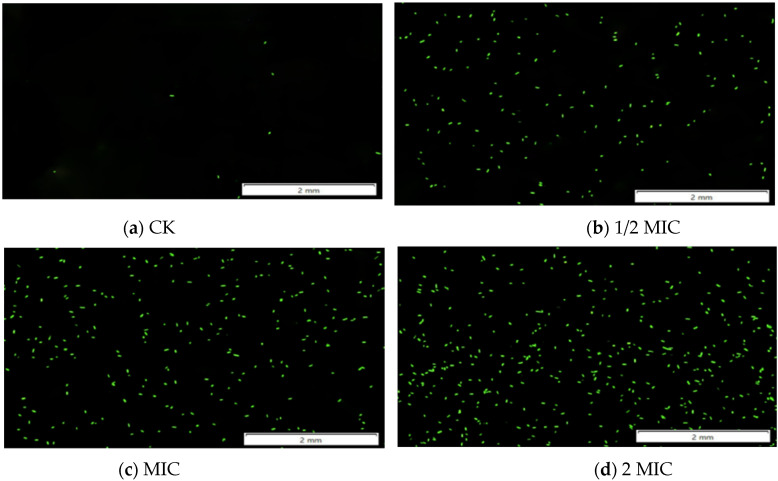
Fluorescence microscope observation of the effect of CEO on ROS in *S. enteritidis*.

**Figure 4 foods-11-02234-f004:**
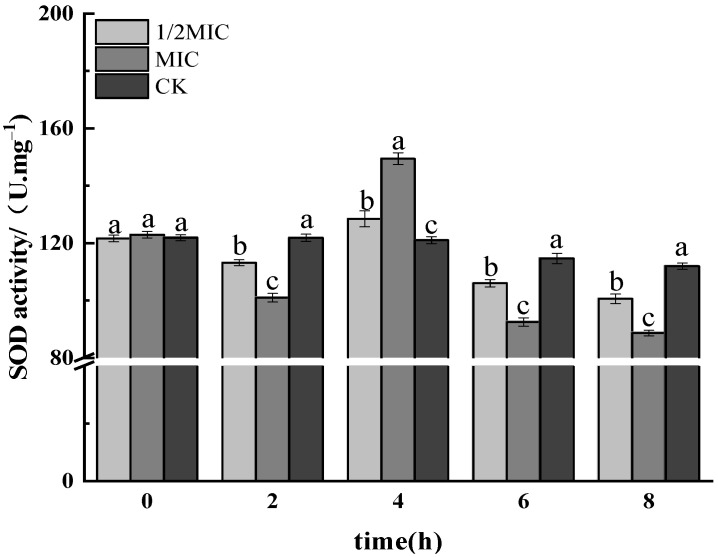
Effect on the enzyme activity of SOD in *Salmonella enteritidis* by CEO (note: a–c indicate significant differences (*p* < 0.05) among 1/2 MIC, MIC, and 2 MIC treatments).

**Figure 5 foods-11-02234-f005:**
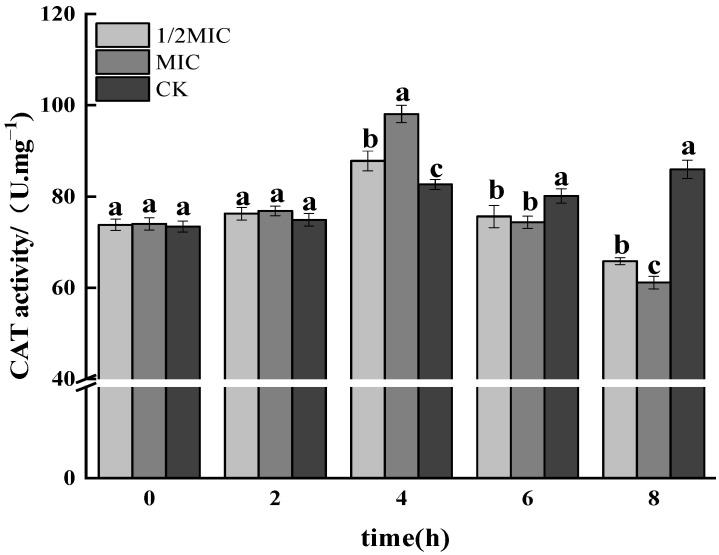
Effect on the enzyme activity of CAT in *Salmonella enteritidis* by CEO (note: a–c indicate significant differences (*p* < 0.05) among 1/2 MIC, MIC, and 2 MIC treatments).

**Figure 6 foods-11-02234-f006:**
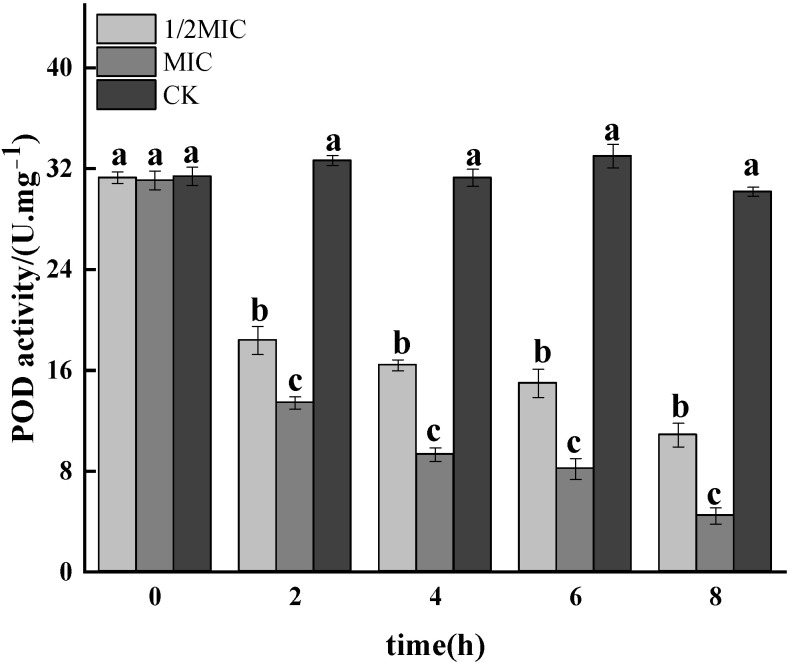
Effect on the enzyme activity of POD in *Salmonella enteritidis* by CEO. (note: a–c indicate significant differences (*p* < 0.05) among 1/2 MIC, MIC, and 2 MIC treatments).

**Figure 7 foods-11-02234-f007:**
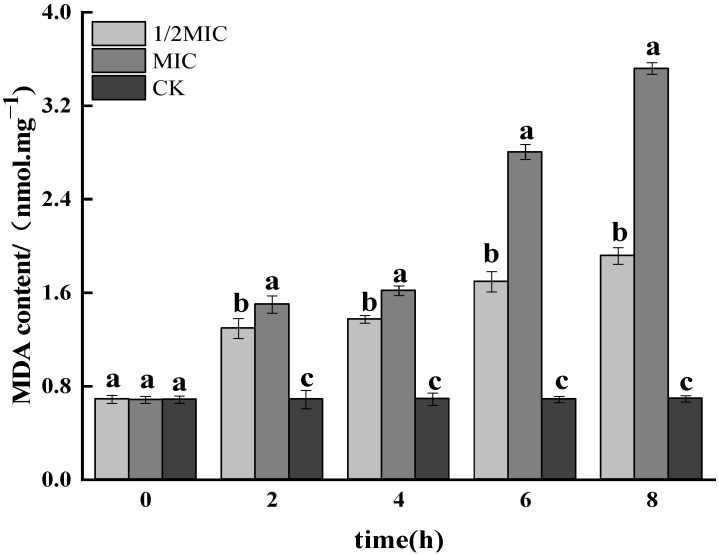
Effect on the concentration of MDA in *Salmonella enteritidis* by CEO (note: a–c indicate significant differences (*p* < 0.05) among 1/2 MIC, MIC, and 2 MIC treatments).

**Figure 8 foods-11-02234-f008:**
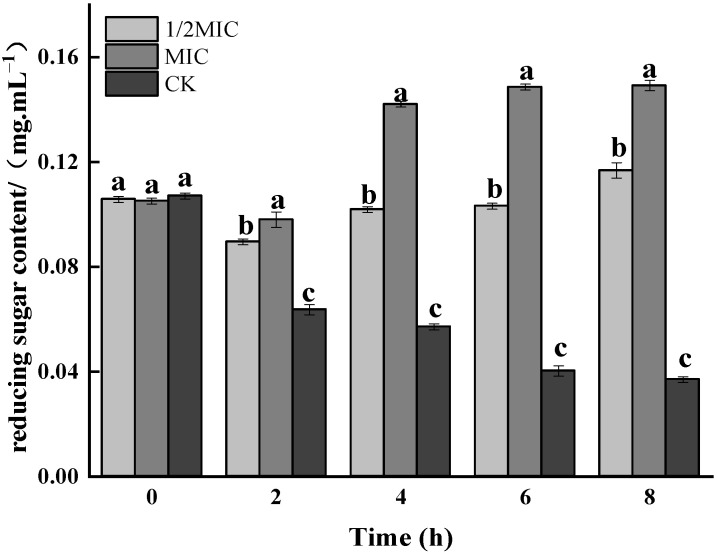
Effects of CEO treatment on protein carbonylation content in *S. enteritidis* (note: a–c indicate significant differences (*p* < 0.05) among 1/2MIC, MIC, 2 MIC treatments).

**Figure 9 foods-11-02234-f009:**
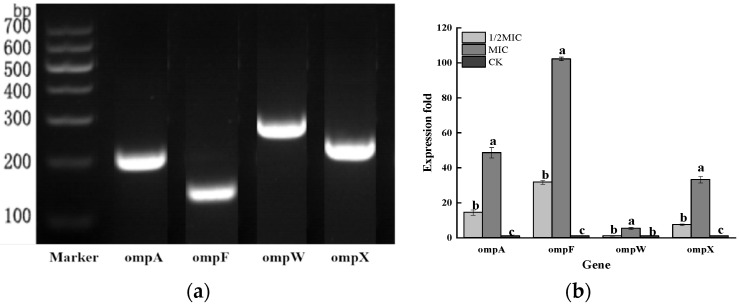
Transcriptional analysis of the membrane protein genes in *Salmonella enteritidis* after CEO treatment (note: a–c indicate significant differences (*p* < 0.05) among 1/2 MIC, MIC, and 2 MIC treatments).

## Data Availability

The data presented in this study are available on request from the corresponding author.

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
