# Peer review of "Effects of Cinnamon Essential Oil on Oxidative Damage and Outer Membrane Protein Genes of Salmonella enteritidis Cells"

_foods, 2022, doi:10.3390/foods11152234_

Round 1
Reviewer 1 Report
The authors of the manuscript “Effects of Cinnamon Essential Oil on Oxidative Damage and Outer Membrane Protein Genes of Salmonella Enteritidis Cells” present an interesting work on the antibacterial mechanism of cinnamon essential oil (CEO). The authors study the effect of CEO on Salmonella enteritidis cells at two different levels: 1) oxidative damage and 2) gene expression of outer membrane proteins.
The manuscript is clearly written and the results are interesting but contains some flaws that need to be improved.
The manuscript does not show line numbers, the peer review is organized on chapter basis.
Abstract
Line 2: authors wrote “Salmonella enteritidis”, please use italic format.
Line 6: authors wrote “Salmonella Enteritidis”, it should be written in italic and lowercase for “enteritidis”.
Line 22: authors wrote “ S. enteritidis”, please use italic format.
Page 1: authors wrote “S.enteritidis” (more or less 8 times), please insert a space in the scientific name, e.g. “S. enteritidis”.
Keywords
Authors wrote “Salmonella enteritidis”, please use italic format.
Introduction
Page 2: authors do not write the reference [2]. Please, if you do not need it anymore… delete it in Reference chapter and organize a new numbered reference list .
Page 2, line 14: please move “Cinnamon essential oil (CEO)” and completing “CEO” at line 10.
Page 2, lines 15-22: authors wrote “Salmonella typhimurium, Listeria monocytogenes, Clostridium oxysporum, pseudomonas aeruginosa”, please use italic format. About the last name, “pseudomonas” should be written with a capitol P.
Page 2, line 27: please insert a space before references “[14-15]”. It should be “… body [14-15]”.
Page 2: authors wrote “S.enteritidis” (several times), please insert a space in the scientific name, e.g. “S. enteritidis”.
Page 2: authors wrote MDA, please may you write the full name? It has been cited 2 times only as MDA.
Materials and Methods
2.1. Materials: see abstract line 6.
2.2.2. O2-• Free Radical Determination: please write OD530 as OD530 (same format of other assays)
2.2.3. Fluorescence microscopy to observe changes in ROS content: Please, use full name for DCFH-DA (first time cited).
2.3.2. CAT, POD and MDA activitydetermination: please, insert a space after activity. Please, insert a space after CAT kit and 420 nm.
2.4. Determination of protein carbonylation: authors wrote DNPH, please may you write the full name? It is first time cited.
2.5.1. PCR amplification: Please, see abstract page 1.
2.5.3. Preparation of S.enteritidis cDNA: please use space and italic format for both cases.
Results and Analysis: please use space and italic format for “S.enteritidis” cases in the chapter…
Figure 1: please, change title of Y axis as “relative fluorescence intensity”, X axis as “time (h)”. It is necessary to explain the letters and the statystical analysis on the figure legend. Better explain X axis on text, is it “time” from treatment or “time” of treatment?
3.1.2. Determination of O2-· free radicals: authors write “Various metabolic processes…”, “Various” shold be written as lowercase.
Figure 2: Letters and X axis as figure 1. Y axis shoudl be written as “O2- nmol x L-1”.
3.1.3. Fluorescence microscopy to observe the changes of ROS: (DCFH) please use full name, you did not do before. Please, explain why only in this case you use 2 MIC.
Figure 3: Authors lack figure 3 and put figure 4 without any description on text... please fix this situation...
Figure 4: See the other figure for X axis, statistical analysis and legend. About Y axis, “specific activity” is U x mg-1. Please, change it
Figure 5 and 6: See figure 4.
Figure 7 and 8: about X axis, legend and statistic follow previous indications. About Y axis “nmol x mg-1” and µmol x mg-1” respectively.
Figure 9: Please use capital letter for distinguish a “panel A” and a “panel B”. Try to use the same format for X and Y axis in both panels
Discussion
Please, insert a space before the reference number. This is valid for each reference of this chapter.
Please use space and italic format for “S.enteritidis” cases in the chapter…
Authors should write in Italic some cited specie names (Candida albicans, Vibrio parahaemolyticus, S. enteritidis).
Authors should consider to write a new sub paragraph about the role of Omp genes (proteins).
Authors should also consider to write at leat 1 sub paragraph about essential oil toxicity on animal (or mammalian) cells. Basically essential oil could enter in food chain if used against S. enteritidis.
References
Reference number 2 is not cited on text.
Please, write in right format the references number 24, 26, 41, 48 and 53.
Author Response
Dear reviewers and editor:
Thank you so much for your email and advices about our paper on Foods entitled “Effects of Cinnamon Essential Oil on OxidativeDamage and Outer Membrane Protein Genes of Salmonella Enteritidis Cells” (ID: foods-1806897). Your valuable comments and suggestions make an improvement for our manuscript and also provide a good guidance to our research. We have reviewed your comments line by line and revised accordingly. The main corrections in the manuscript and the responds to the reviewers’ and editor's comments are as following:
REVIEWER REPORT:
Referee: 1
Comments and Suggestions for Authors:The authors of the manuscript “Effects of Cinnamon Essential Oil on Oxidative Damage and Outer Membrane Protein Genes of Salmonella Enteritidis Cells” present an interesting work on the antibacterial mechanism of cinnamon essential oil (CEO). The authors study the effect of CEO on Salmonella enteritidis cells at two different levels: 1) oxidative damage and 2) gene expression of outer membrane proteins.
The manuscript is clearly written and the results are interesting but contains some flaws that need to be improved.
The manuscript does not show line numbers, the peer review is organized on chapter basis.
Abstract
Question:Line 2: authors wrote “Salmonella enteritidis”, please use italic format.
Response:Thanks for your kind advice. We have corrected it according to your suggestion. Namely, change "Salmonella enteritidis" to "Salmonella enteritidis" see the red in line 2 of abstract,on page 1.
Question:Line 6: authors wrote “Salmonella Enteritidis”, it should be written in italic and lowercase for “enteritidis”.
Response:Thanks for your kind advice. We have corrected it according to your suggestion. Namely, change "Salmonella enteritidis." to "Salmonella enteritidis." , The word “Enteritidis” had been changed lowercase“enteritidis” see the red in line 6 of abstract,on page 1.
Question:Line 22: authors wrote “ S. enteritidis”, please use italic format.
Response:Thanks for your kind advice. We have corrected it according to your suggestion. Namely, change "S. enteritidis" to "S. enteritidis" , see the red in line 22 of abstract,on page 1. All occurrences of "S. enteritidis" in the text are changed to italics.
Question:Page 1: authors wrote “S.enteritidis” (more or less 8 times), please insert a space in the scientific name, e.g. “S. enteritidis”.
Response:Thanks for your kind advice. We have corrected it according to your suggestion. Namely, change "S.enteritidis" to "S. enteritidis" , see the red of abstract,on page 1. All occurrences of "S. enteritidis" in the text are insertt a space.
Keywords
Question:Authors wrote “Salmonella enteritidis”, please use italic format.
Response:Thanks for your kind advice. We have corrected it according to your suggestion. Namely, change "SSalmonella enteritidis" to "Salmonella enteritidis" , see the red of keywords,on page 1. All occurrences of "Salmonella enteritidis" in the text are changed to italics format.
Introduction
Question:Page 2: authors do not write the reference [2]. Please, if you do not need it anymore… delete it in Reference chapter and organize a new numbered reference list .
Response:Thanks for your kind advice. Sorry for the error caused by our carelessness. Literature 2 has been marked in the paper. We have corrected it according to your suggestion. see the red in line 2 on page 2.
Question:Page 2, line 14: please move “Cinnamon essential oil (CEO)” and completing “CEO” at line 10.
Response:Thanks for your kind advice. Sorry for the confusion caused by our carelessness. We have corrected it according to your suggestion. We have moved “Cinnamon essential oil (CEO)” and completing “CEO” at line 10.see the red in line 10 on page 2.
Question:Page 2, lines 15-22: authors wrote “Salmonella typhimurium, Listeria monocytogenes, Clostridium oxysporum, pseudomonas aeruginosa”, please use italic format. About the last name, “pseudomonas” should be written with a capitol P.
Response:Thanks for your kind advice. We have corrected it according to your suggestion. Namely, change“Salmonella typhimurium, Listeria monocytogenes, Clostridium oxysporum, pseudomonas aeruginosa” to “Salmonella typhimurium, Listeria monocytogenes, Clostridium oxysporum, pseudomonas aeruginosa” , the last name, “pseudomonas” have been written with a capitol P , see the red lines 14-23,on page 2. All occurrences of “Salmonella typhimurium, Listeria monocytogenes, Clostridium oxysporum, pseudomonas aeruginosa” in the text are changed to italics format.
Question:Page 2, line 27: please insert a space before references “[14-15]”. It should be “… body [14-15]”.
Response:Thanks for your kind advice. We have corrected it according to your suggestion. Namely, a space have been inserted before references “[14-15]”,see the red lines 28-29, on page 2.
Question:Page 2: authors wrote “S.enteritidis” (several times), please insert a space in the scientific name, e.g. “S. enteritidis”.
Response:Thanks for your kind advice. We have corrected it according to your suggestion. Namely, a space have been inserted in the scientific name, e.g. “S. enteritidis”, see the red in full text.
Question:Page 2: authors wrote MDA, please may you write the full name? It has been cited 2 times only as MDA.
Response:Thanks for your kind advice. I am sorry, we have not corrected it according to your suggestion. Because, MDA has been cited 20 times, and MDA have not been writeen the full name- “malondialdehyde”, see the red line 39, on page 3.
Materials and Methods
Question:2.1. Materials: see abstract line 6.
Response:Thanks for your kind advice. We have corrected it according to your suggestion. Namely, change "Salmonella enteritidis" to "Salmonella enteritidis" see the red in line 2 of 2.1. Materials ,on page 3.
Question:2.2.2. O2-• Free Radical Determination: please write OD530 as OD530 (same format of other assays)
Response:Thanks for your kind advice. We have corrected it according to your suggestion. Namely, write OD530 as OD530 (same format of other assays), see the red in line 7 of 2.2.2. O2-• Free Radical Determination,on page 3.
Question:2.2.3. Fluorescence microscopy to observe changes in ROS content: Please, use full name for DCFH-DA (first time cited).
Response:Thanks for your kind advice. We have corrected it according to your suggestion. Namely, DCFH-DA have been changed full name (2’,7’-Dichlorodihydrofluorescein diacetate), see the red in line 5 of 2.2.3. Fluorescence microscopy to observe changes in ROS content,on page 4.
Question:2.3.2. CAT, POD and MDA activitydetermination: please, insert a space after activity. Please, insert a space after CAT kit and 420 nm.
Response:Thanks for your kind advice. We have corrected it according to your suggestion. Namely, a space have been inserted after activity, CAT kit and 420 nm., see the red in lines 2-3 of 2.3.2. CAT, POD and MDA activitydetermination, on page 4.
Question:2.4. Determination of protein carbonylation: authors wrote DNPH, please may you write the full name? It is first time cited.
Response:Thanks for your kind advice. We have corrected it according to your suggestion. Namely, DNPH have been written the full name, see the red in line 9 of 2.4. Determination of protein carbonylation, on page 4.
Question:2.5.1. PCR amplification: Please, see abstract page 1.
Response:Thanks for your kind advice. We have corrected it according to your suggestion. Namely, according to abstract page 1, “Salmonella Enteritidis” have been changed italic format see the red in line 1 of 2.5.1. PCR amplification, on page 4.
Question:2.5.3. Preparation of S. enteritidis cDNA: please use space and italic format for both cases.
Response:Thanks for your kind advice. We have corrected it according to your suggestion. Namely, space and italic format have been used, see the red in title and line 1 of 2.5.3. Preparation of S. enteritidis cDNA, on page 4.
Results and Analysis
Question: please use space and italic format for “S.enteritidis” cases in the chapter…
Response:Thanks for your kind advice. We have corrected it according to your suggestion. Namely, space and italic format have been used for “S.enteritidis” cases in the chapter, see the red in the chapter of Results and Analysis, on page 5-12. And full text have been changed about same question.
Question:Figure 1: please, change title of Y axis as “relative fluorescence intensity”, X axis as “time (h)”. It is necessary to explain the letters and the statystical analysis on the figure legend. Better explain X axis on text, is it “time” from treatment or “time” of treatment?
Response:Thanks for your kind advice. We have corrected it according to your suggestion. Namely, title of Y axis and X axis have been changed as “relative fluorescence intensity” and “time (h)”. , see Figure 1 , on page 5. It has been explained of the letters and the statystical analysis on the figure legend, see figure 1, and all of same question have been explianed see figure 1-9
Question:3.1.2. Determination of O2-· free radicals: authors write “Various metabolic processes…”, “Various” shold be written as lowercase.
Response:Thanks for your kind advice. We have corrected it according to your suggestion. Namely, “Various” have been written as lowercase “various” , see the red in line 6 of 3.1.2. Determination of O2-· free radicals, on page 5.
Question:Figure 2: Letters and X axis as figure 1. Y axis shoudl be written as “O2- nmol x L-1”.
Response:Thanks for your kind advice. We have corrected it according to your suggestion. Namely, title of Y axis and X axis have been changed as “O2- nmol x L-1” and “time (h)”. , see Figure 2 , on page 6. It has been explained of the letters and the statystical analysis on the figure legend, see figure 1, and all of same question have been explianed see figure 1-9
Question:3.1.3. Fluorescence microscopy to observe the changes of ROS: (DCFH) please use full name, you did not do before. Please, explain why only in this case you use 2 MIC.
Response:Thanks for your kind advice. We have corrected it according to your suggestion. Namely, DCFH have been used full name (2’,7’-Dichlorodihydrofluorescein diacetate), see red in line 2 of 3.1.3. Fluorescence microscopy to observe the changes of ROS, on page 6. In this case 2 MIC are been used, only because 4 pictures put together looks like not ugly. If it is unnecessary,we can delete it.
Question:Figure 3: Authors lack figure 3 and put figure 4 without any description on text... please fix this situation...
Response:Thanks for your kind advice. We have corrected it according to your suggestion. The figure 3 locate on page 7 in text, at the same time, the figure 3 shows as follow. The describe of figure 4 locate on page 7 in text, at the same time. The describe of figure 4 shows 3.1.4. SOD activity assay on page 7 in text. And the describe of figure 4 also shows after figure 3 on this page.
Figure 3. Fluorescence microscope observation of the effect of CEO on ROS in S. enteritidis.
3.1.4. SOD activity assay
SOD was one of the key protective enzymes in bacteria, which could scavenge O2-• and improve the tolerance of bacteria to ROS, so as to protect the cells from oxidative damage [31]. The results showed that after Salmonella enteritidis was treated with different concentrations of CEO for 2 h, the SOD activity decreased to varying degrees. At 4 h, the SOD activity increased rapidly, and the higher the CEO concentration, the greater the increase. Among them, the SOD activity of the MIC treatment group reached the maximum at 4 h, which was 149.16 U.mgprot−1; at 8 h, the SOD activity decreased significantly, and the higher the CEO concentration, the more significant the SOD activity decreased (P<0.05), while The control group showed no significant change. Therefore, it was shown that CEO treatment had a significant effect on the SOD activity of S.enteritidis, and the SOD activity decreased, indicating that its protection to Salmonella enteritidis itself decreased, and it was easily attacked by free radicals and autolysed.
Question:Figure 4: See the other figure for X axis, statistical analysis and legend. About Y axis, “specific activity” is U x mg-1. Please, change it
Response:Thanks for your kind advice. We have corrected it according to your suggestion. Namely, title of Y axis and X axis have been changed as “U x mg-1” and “time (h)”. , see figure 4 , on page 7. It has been explained of the letters and the statystical analysis on the figure legend, see figure 1, and all of same question have been explianed see figure 1-9.
Question:Figure 5 and 6: See figure 4.
Response:Thanks for your kind advice. We have corrected it according to your suggestion. Namely, title of Y axis and X axis have been changed as “U x mg-1” and “time (h)”. , see figure 5 and 6 , on page 8. It has been explained of the letters and the statystical analysis on the figure legend, see figure 1, and all of same question have been explianed see figure 1-9.
Question:Figure 7 and 8: about X axis, legend and statistic follow previous indications. About Y axis “nmol xmg-1” and µmol x mg-1” respectively.
Response:Thanks for your kind advice. We have corrected it according to your suggestion. Namely, title of Y axis and X axis have been changed as “U x mg-1” and “time (h)” , see figure 7 and 8, on page 9 and 10. About X axis, legend and statistic have changed follow previous indications. It has been explained of the letters and the statystical analysis on the figure legend, see figure 1, and all of same question have been explianed see figure 1-9.
Question:Figure 9: Please use capital letter for distinguish a “panel A” and a “panel B”. Try to use the same format for X and Y axis in both panels
Response:Thanks for your kind advice. We have corrected it according to your suggestion. Namely, capital letter for distinguish a “panel A” and a “panel B” have been used , see figure 9 , on page 11. It has been explained of the letters and the statystical analysis on the figure legend, see figure 1, and all of same question have been explianed see figure 1-9.
Discussion
Question:Please, insert a space before the reference number. This is valid for each reference of this chapter.
Response:Thanks for your kind advice. We have corrected it according to your suggestion. Namely, space before the reference number have been inserted , see part of discussion , on page 11-12.
Question:Please use space and italic format for “S.enteritidis” cases in the chapter…
Response:Thanks for your kind advice. We have corrected it according to your suggestion. Namely, space and italic format have been used for “S.enteritidis” cases in the chapter, see the red “S.enteritidis” in the chapter of discussion, on page 11-12. And full text have been changed about same question.
Question:Authors should write in Italic some cited specie names (Candida albicans, Vibrio parahaemolyticus, S. enteritidis).
Response:Thanks for your kind advice. We have corrected it according to your suggestion. Namely, italic some cited specie names (Candida albicans, Vibrio parahaemolyticus, S. enteritidis)
have been written , see the red in the chapter of discussion, on page 11-12. And full text have been changed about same question.
Question:Authors should consider to write a new sub paragraph about the role of Omp genes (proteins).
Response:Thanks for your kind advice. We have corrected it according to your suggestion. Namely,paragraph about the role of Omp genes (proteins) have been written , see the red in the chapter of discussion, on page 11-12.
Question:Authors should also consider to write at leat 1 sub paragraph about essential oil toxicity on animal (or mammalian) cells. Basically essential oil could enter in food chain if used against S. enteritidis.
Response:Thanks for your kind advice. We have corrected it according to your suggestion. Namely, paragraph about essential oil toxicity on animal (or mammalian) cells have been written , see the blue back in the chapter of discussion, on page 13.
References
Question:Reference number 2 is not cited on text.
Response:Thanks for your kind advice. Sorry for the error caused by our carelessness. Literature 2 has been marked in the paper. We have corrected it according to your suggestion. see the red in line 2 on page 2.
Question:Please, write in right format the references number 24, 26, 41, 48 and 53.
Response:Thanks for your kind advice. We have corrected it according to your suggestion. Namely, right format the references number 24, 26, 41, 48 and 53 have been written 34,36,51,58 and 63, see the red in the chapter of References, on page 14-15.

Reviewer 2 Report
Where the references of these methods

Author Response
Thanks for you kind advices, we have corrected according to your suggestion. Because pictures of question are not been showed on this page, And see PDF for details of referee 2.
Question:Specific comments are shown on the right side of the figure below
Response:Thanks for your kind advice. We have corrected it according to your suggestion. Namely, see the blue words, on page 1.
Question:Specific comments are shown on the right side of the figure below
Response:Thanks for your kind advice. We have corrected it according to your suggestion. Namely, see the blue words and green back , on page 2. And the method of extration CEO have been writeen on page 3 of 2.1. Materials, see blue words.
Question:Specific comments are shown on the right side of the figure below
Response:Thanks for your kind advice. We have corrected it according to your suggestion. Namely, “activity determination” has been corrected, see the green back of 2.3.2. CAT, POD and MDA activity determination on page 4. The references of these method have been added, which was 30, 31, 33, 34.The table has been insert in supporting data.The conclution has been rewriten, see green back on page 12. The table 1 has been put supporting date.

Reviewer 3 Report
The manuscript addresses the study of Cinnamomum essential oil against Salmonella enteritis. The work needs some clarification for better understanding and reliability of the results.
The manuscript lacks an explanation of the purposes and tests performed. First, the genus Cinnamomum has about 300 species distributed throughout the world. Which species was chosen for the study? Even if it has been purchased, it is necessary to identify the species, as many of them are used as spices. However, for some, cinnamaldehyde is not the major compound in the volatile oil. In the introduction, the authors mention two references to talk about essential oil chemistry, but they are different species of Cinnamomum. So which species is the study?
Regarding the objectives, the problem with Salmonella and with ROS is explained. Add explanation for working with Omp genes (OmpA, OmpF, OmpW and OmpX).
Authors must review the entire manuscript considering the correct spelling of scientific names. There are many errors, especially with the italic writing and uppercase and lowercase letters of the species of microorganisms mentioned throughout the text.
In material and methods, the authors comment that "The cDNA obtained in the previous experiment was used as the template". What previous experiment was this?
In the ROS test, what is the control group? What is CK? Who is positive control? How was the essential oil sample diluted, since it does not solubilize in an aqueous medium? What is the oil concentration relative to 1/2 MIC and MIC?
In O2-• Free Radical Determination, what is 150 r/min?
Figures need to be clearer. Add caption with the meaning of the abbreviations. Mention the statistic in the legend. How many repetitions were performed for each experiment?
Author Response
See PDF for details

Round 2
Reviewer 1 Report
The authors have addressed comments/changes as requested. The manuscript is ready to be published in Foods. But… before, authors have to check the italic format of specie names (still) on new added paragraph of discussion (blu higlighted) and green highlighted part of the conclusion.
Reviewer 2 Report
I think the manuscript is ready for publication in the present form where the authors made overall required revisions
Reviewer 3 Report
The manuscript entitled "Effects of Cinnamon Essential Oil on Oxidative Damage and Outer Membrane Protein Genes of Salmonella Enteritidis Cells" aims to provide a theoretical basis for the development of new natural food preservatives and the prevention and control of Salmonella enteritidis. The requested corrections have been met. So, the manuscript is ok to be accepted.